# Association between Different Types of Plant-Based Diets and Dyslipidemia in Middle-Aged and Elderly Chinese Participants

**DOI:** 10.3390/nu15010230

**Published:** 2023-01-02

**Authors:** Lu Wang, Yuanyuan Li, Yan Liu, Huanwen Zhang, Tingting Qiao, Lei Chu, Tao Luo, Zewen Zhang, Jianghong Dai

**Affiliations:** 1School of Public Health, Xinjiang Medical University, 393 Xin Medical Road, Urumqi 830011, China; 2Department of Nutrition, School of Public Health, Sun Yat-sen University, Guangzhou 510000, China

**Keywords:** plant-based dietary patterns, dyslipidemia, plant-based-food quality, middle-aged and elderly Chinese

## Abstract

Plant-based dietary patterns may reduce the risk of dyslipidemia. However, not all plant-based foods are beneficial, and limited data exist for the Chinese population. We investigated the association between different plant-based dietary indices and the risk of dyslipidemia in a Chinese middle-aged and elderly population. The study participants (n = 4096) consisted of adults between 35 and 74 years of age from Xinjiang, China. Dietary consumption of the study participants was evaluated using a semi-quantitative food-frequency questionnaire (FFQ). Three different plant-based dietary indices were calculated using data from dietary surveys, including overall plant-based diet index (PDI), healthy plant-based diet index (hPDI), and unhealthy plant-based diet index (uPDI). Based on these indices, we created an adjusted plant-based diet index (aPDI) based on the Xinjiang population actual dietary behavior and health effects of food. We measured the levels of total cholesterol, triglyceride, LDL-C, and HDL-C in the blood of the study participants. We used multivariable logistic regression and restricted cubic spline to analyze the relationship between plant-based diets and dyslipidemia. The findings showed that 36.6% of the participants had dyslipidemia. Higher PDI adherence was related to lower odds of dyslipidemia (Q3 vs. Q1, OR: 0.780, 95% CI: 0.641–0.949; Q4 vs. Q1, OR: 0.799, 95% CI: 0.659–0.970). High aPDI was related to lower odds of dyslipidemia (Q4 vs. Q1, OR: 0.770, 95% CI: 0.628–0.945; Q5 vs. Q1, OR: 0.748, 95% CI: 0.607–0.921). High scores for PDI, hPDI, and aPDI were all related to a reduced risk of low HDL-C (OR: 0.638, 95% CI: 0.491–0.823; OR: 0.661, 95% CI: 0.502–0.870; OR: 0.580, 95% CI: 0.443–0.758). Conversely, a high uPDI score was associated with an increased risk of low HDL-C (OR: 1.349, 95% CI: 1.046–1.740). There was no non-linear relationship between PDI, hPDI, uPDI, and aPDI and the risk of different types of dyslipidemia. Plant-based dietary indices are related to specific types of dyslipidemia risk. Appropriately increasing the consumption of plant-based foods while improving the quality of plant-based dietary patterns is critical for the prevention of dyslipidemia, especially low HDL-C, in the population.

## 1. Introduction

Dyslipidemia, which is closely related to atherosclerosis, is a key modifiable factor in the development of several cardiovascular diseases [1,2,3]. Among the Chinese population aged 35 to 75 years, the prevalence of dyslipidemia is 33.8% [4]. With the global prevalence of dyslipidemia, dyslipidemia has gradually become a public health concern that poses a threat to human health and causes a huge disease burden [5,6,7].

The factors affecting dyslipidemia are complex. A healthy diet is an effective way to prevent and improve dyslipidemia [8,9]. Previous studies have shown that plant-based foods such as fresh vegetables and fruits have health-promoting properties [10]. The quantity of plant-based foods is strongly related to a reduction in blood lipid levels [11,12]. Even though vegan or vegetarian diets have been recognized as a relatively healthy dietary pattern and are gradually being accepted by some members of the public, not all plant-based foods are beneficial to health [13,14]. Unhealthy plant-derived foods such as sugar-sweetened beverages are related to a high risk of dyslipidemia [15].

Satija et al. created the PDI, hPDI, and uPDI based on the degrees of healthy plant-based foods. These indices provide a comprehensive and holistic analysis of plant-based dietary patterns [16]. Some studies have reported that higher PDI and/or hPDI are related to a low risk of coronary heart disease (CHD), type 2 diabetes mellitus(T2DM), and obesity, while individuals with high uPDI scores are related to a high risk of these chronic diseases [17,18,19]. However, these indices have some limitations. For example, these indices only consider the health effects of plant-based foods; however, not all animal foods are harmful to human health. Meat, seafood, dairy products, and other animal foods are rich in protein, amino acids, vitamins, and trace elements necessary for growth and development. An adequate consumption of these animal foods has considerable effects on human growth and development and physiological regulation [20,21,22,23]. To overcome this limitation, we created aPDI as a complement for a more comprehensive assessment of dietary behavior habits dominated by plant foods. aPDI was created based on the classification criteria of PDI, considering the intake of partially healthy animal-based foods. In addition, aPDI was established with some adjustments to the food types and food score directions according to the dietary habits of Xinjiang residents in China, which is more suitable for evaluating the dietary intake of Xinjiang residents. 

In parallel with a growing concern on climate changes, plant-based dietary patterns and their effects on health are receiving increasing attention from researchers. Understanding the relationship between plant-based dietary patterns and dyslipidemia may help improve the lipid profiles and thereby reduce the risk of chronic diseases. Nevertheless, studies on the association between plant-based dietary patterns and lipid profiles have shown mixed results [24,25,26]. China is a vast country, and the dietary patterns of the population are diverse and relatively complex. Xinjiang, as a multi-ethnic population cluster, has developed unique dietary habits that deserve attention. In this study, we used baseline data from the Xinjiang Multiethnic Cohort (XMC) study to explore whether plant-based dietary patterns are related to dyslipidemia in Chinese residents.

## 2. Materials and Methods

### 2.1. Study Population

The XMC study was a prospective cohort study involving participants from Urumqi, Hotan, and Yili in Xinjiang, China. The XMC study investigated the potential causal relationships between factors such as geography, dietary behavior habits, and lifestyle and health outcomes of the Xinjiang population [27]. Our study analyzed baseline survey data from Yili, which included 8011 residents aged 35–74 years. The participants were enrolled at three township health service centers in Huocheng County, Yili, from December 2018 to May 2019, and follow-up has not been completed yet. The study was approved by the Ethics Committee of the Xinjiang Uygur Autonomous Region Institute of Traditional Chinese Medicine, and written informed consent was signed by each participant. Participants with missing information on lipid glucose variables (n = 2445), missing dietary information for calculating PDI (n = 1384), and missing covariate information (n = 86) were excluded. The final sample size included in the analysis was 4096.

### 2.2. Physical Examination and Biochemical Assays

Participants were measured twice for weight, height, and waist circumference by trained nurses or physicians. We measured blood pressure using a medical arm electronic sphygmomanometer (two measurements per participant, 5–10 min apart), calculated from an average of two measurements. Nurses collected fasting venous blood from the participants in the morning. Indicators such as blood glucose and blood lipids were determined at the nearest township health center to the survey location. 

### 2.3. Definition of Diagnostic Criteria and Related Indicators

Diagnostic criteria for dyslipidemia [28]: (1) hypercholesteremia: total cholesterol (TC) ≥ 6.20 mmol/L; (2) hypertriglyceridemia: triglycerides (TG) ≥ 2.30 mmol/L; (3) high low-density lipoprotein-cholesterol (LDL-C): LDL-C ≥ 4.10 mmol/L; (4) low high-density lipoprotein-cholesterol (HDL-C): HDL-C < 1.00 mmol/L; (5) self-reported history of dyslipidemia. The participants were considered to be dyslipidemic if they met any of the above criteria. 

### 2.4. Assessment of PDI Score

The FFQ of this study was created based on the questionnaire designed by the XMC project team and modified according to the dietary characteristics of Yili residents. Trained investigators used a face-to-face survey to obtain information from participants on the frequency of food consumption (daily, 4–6 times/week, 1–3 times/week, 1–3 times/month, no or very little food) and the average intake per occasion in the past year. 

We calculated PDI, hPDI, and uPDI as previously reported [16]. Briefly, the frequencies of consumption and intake of 127 food items were converted to daily consumption and classified into 18 food groups, and the 18 food groups were further subdivided into three major categories according to the PDI calculation rules: healthy plant foods (i.e., whole grains, fruits, vegetables, nuts, legumes, vegetable oils, and tea or coffee), unhealthy plant foods (i.e., fruit juices, refined grains, potatoes, sugary drinks, and desserts), and animal foods (i.e., animal fats, dairy products, eggs, fish or seafood, poultry or red meat, and other animal foods). Each food group was divided into Q1 to Q5 groups based on the quintile of average daily intake for each food group. For PDI, participants were given 5 scores in each plant food group for which they were in the highest quintile of consumption, and 1 score in the lowest consumption quintile (positive scores). Conversely, participants received 1 score for each animal food group in the highest consumption quintile and 5 scores in the lowest consumption quintile (negative scores). For hPDI, positive scores were used for healthy plant foods, and negative scores were assigned to unhealthy plant foods and animal foods. Finally, for uPDI, positive scores were assigned to unhealthy plant foods, and negative scores were assigned to healthy plant foods and animal foods. Moreover, we classified the 18 food groups based on the dietary patterns of Xinjiang residents and the health effects of foods. While keeping the plant-based food classification unchanged, we divided animal-based foods into healthy animal foods (i.e., dairy products, eggs, fish or seafood, and poultry or red meat) and unhealthy animal foods (i.e., animal fats and other animal foods). For aPDI, we assigned positive scores to healthy plant foods and healthy animal foods and negative scores to unhealthy plant foods and unhealthy animal foods. The scores of the 18 food components were added up to the total score.

### 2.5. Assessment of Covariates

The survey was conducted by trained and qualified surveyors using a face-to-face approach. Specifically, the surveyors used a structured questionnaire to collect basic information (e.g., sex, age, ethnicity, education level, occupation, annual per capita household income, and property consumption) and health information (alcohol consumption, physical activity, and history of chronic diseases) of the participants. BMI was calculated as weight divided by standing height squared (kg/m^2^). Atherogenic index of plasma (AIP) was calculated as log10 (TG/HDL-C). To assess socioeconomic status (SES), this study used four dimensions of education level, occupation, annual per capita household income, and property. We assigned scores 0, 1, 2, and 3 for illiteracy, elementary school, junior high school, and high school and above, respectively; scores 0, 1, 2, and 3 for jobless, farmer, employee, and private owner or professional technician, respectively; and scores 0, 1, and 2 for per capita annual household income below RMB 10,000, between RMB 10,000 and RMB 35,000, and above RMB 35,000, respectively; property included house, bathroom, car, motorcycle, computer, internet access, smartphones, and travel in the past five years, we assigned scores of 0, 1, and 2 for <2 properties, 2 to 3 properties, and >3 properties, respectively. The overall SES score was calculated by summing the scores of the four areas, with a maximum total score of 10, where 1 to 3, 4 to 7, and 8 to 10 are defined as low, medium, and high, respectively. 

### 2.6. Statistical Analysis

We used the mean and standard deviation to present continuous data satisfying a normal distribution, the median and interquartile to present continuous data not satisfying a normal distribution, and frequency and proportion to present categorical variables. We applied the Student’s *t*-test for normally distributed continuous data, the Mann–Whitney U test for non-normally distributed continuous data, and χ^2^ tests for categorical data. Multivariate logistic regression models were used to estimate the odds ratio (ORs) and 95% confidence intervals (CIs) for four plant-based dietary index quintiles and different types of dyslipidemia. We further explored the nonlinear association between four plant-based dietary indices and dyslipidemia using restricted cubic splines (RCS) and selected four nodes for curve fitting according to the Akaike information criterion (AIC) optimality principle. For all statistical analyses, we used R 4.1.2. *p* < 0.05 (two-tailed) was statistically significant. Images were generated with GraphPad Prism 8.

## 3. Results

### 3.1. Characteristics of Samples 

Table 1 shows the characteristics of the participants in the study. Out of 4096 participants, 1501 (36.6%) had dyslipidemia. There were 1840 males (44.9%) and 2256 (55.1%) females. The differences in sex, ethnicity, and BMI between the dyslipidemic and normolipidemic groups were significant (*p* < 0.05). Dyslipidemic participants had significantly higher levels of blood glucose and lipid. As expected, dyslipidemic participants had higher fasting plasma glucose (FPG), TC, TG, and LDL-C levels, and lower AIP and HDL-C levels, than normolipidemic participants (*p* < 0.001). Figure 1 shows the differences in blood parameters between the dyslipidemic and normolipidemic groups. 

### 3.2. Correlation between PDI and Dyslipidemia

Table 2 shows the relationship between the four plant-based dietary indices and dyslipidemia. After adjusting for confounding factors, higher PDI was related to lower odds of dyslipidemia (Q3 vs. Q1, OR: 0.780, 95% CI: 0.641–0.949; Q4 vs. Q1, OR: 0.799, 95% CI: 0.659–0.970), and adherence to aPDI was related to lower odds of dyslipidemia (Q4 vs. Q1, OR: 0.770, 95% CI: 0.628–0.945; Q5 vs. Q1, OR: 0.748, 95% CI: 0.607–0.921). Figure 2, Figure 3, Figure 4 and Figure 5 show the association between the four plant-based dietary indices and dyslipidemia and its components. After adjusting for confounders, PDI, hPDI, and aPDI were related to a reduced risk of low HDL-C. Being in the highest quintile of PDI, hPDI, and aPDI was found to reduce the odds of low HDL-C by 36.2%, 33.9%, and 42.0%, respectively, compared to being in the lowest quintile (OR: 0.638, 95% CI: 0.491–0.823; OR: 0.661, 95% CI: 0.502–0.870; OR: 0.580, 95% CI: 0.443–0.758, respectively). Adherence to uPDI was related to increased odds of low HDL-C, and the relationship remained after adjusting for potential confounders. Compared to individuals in the lowest quintile, uPDI in the highest quintile was associated with 34.9% higher odds of low HDL-C (OR: 1.349, 95% CI: 1.046–1.740). The results of the partial correlation analysis between dyslipidemia-related indicators and plant-based diet indices are shown in Appendix A.

In the RCS based on a logistic regression model, none of the nonlinear spline tests was statistically significant (*p* nonlinearity > 0.05), indicating that there was no potential nonlinear association between the four plant-based dietary indices and the risk of dyslipidemia (Appendix A).

## 4. Discussion

In this multi-ethnic population study of Xinjiang middle-aged and elderly, we discovered that higher PDI and aPDI were related to a lower risk of dyslipidemia. However, there was no significant correlation between hPDI and uPDI and dyslipidemia. When the four plant-based dietary indices were further investigated in association with different types of dyslipidemia, we observed that higher PDI, hPDI, and aPDI were related to a reduced risk of low HDL-C, while higher uPDI was related to an increased risk of low HDL-C. These results demonstrate that the consumption of plant-based foods is critical for the prevention of dyslipidemia, especially low HDL-C.

Diet is a key modifiable factor in a variety of chronic diseases. As the effects of plant-based diets on the health of populations are gradually being discovered, there are increasing recommendations from the scientific community to shift to plant-based dietary patterns [29]. According to the EAT–Lancet Commission, the intake of plant-based foods and the limitation of animal-source foods can be effective in improving health [30]. This argument has also been confirmed in several studies. According to the results of several large population-based prospective cohort studies, a high level of compliance with PDI was inversely related to CHD, with this reverse association stronger for hPDI. Additionally, there was a positive association between uPDI and CHD [17]. PDI and hPDI were related to lower odds of T2DM and reduced weight gain. Conversely, uPDI was related to more weight gain [31,32].

While the health effects of plant-based foods are confirmed, the mere consumption of vegetarian foods can also pose certain health risks. Plant-based dietary patterns are low in protein and several essential micronutrients such as vitamins B12 and D and calcium [33]. Hence, the risk of adverse reactions attributed to micronutrient deficiencies should not be underestimated. A prospective cohort study reported that vegetarians have a higher risk of stroke compared to meat eaters [34]. Long-term vegetarianism reduces choline levels in the brain, which can contribute to brain damage [35]. Moreover, the intake of unhealthy plant-based foods with high contents of sugar and low contents of dietary fiber, unsaturated fatty acids, trace minerals, and antioxidants, such as sugary drinks and fruit juices, may cause adverse effects and increase the risk of cancer [36]. Therefore, both the quality of plant foods and the consumption of an appropriate amount of animal foods should be considered. Even though PDI considers both plant- and animal-based food consumption and the quality of plant foods, it fails to distinguish the quality of animal foods.

A study in Korea showed that PDI and hPDI were negatively related to dyslipidemia risk, while uPDI was positively related to dyslipidemia [37]. However, in our study, neither hPDI nor uPDI was significantly associated with dyslipidemia, which may be related to the fact that hPDI and uPDI had negative scores for all animal foods and did not consider the quality of animal foods. Therefore, this study established aPDI by considering the health effects of animal foods and categorizing animal foods into healthy and unhealthy animal foods. Animal foods are a double-edged sword, with health effects on both sides. On the one hand, meat is rich in nutrients, and its proper consumption has positive health effects. Livestock, poultry, eggs, and aquatic products represent good sources of dietary protein, fat, vitamin A, B vitamins, and minerals, and play important roles in the human diet [22,38,39]. On the other hand, the overconsumption of meat increases disease risk [40,41]. Therefore, it is important to consume animal foods in moderation. The Dietary Guidelines for Chinese Residents (2022) [42] recommend that adults have a daily intake of 120–200 g of animal foods such as fish, poultry, meat, and eggs. The recommended intake is equivalent to 300 g of milk per day. In this study, the average intake of meat and eggs was 107 g, and the average intake of dairy products was 151 g. There was no overconsumption of animal foods among the participants. In this study, poultry, red meat, fish and seafood, eggs, and dairy products were defined as healthy animal foods and were assigned positive scores in the calculation of aPDI. Classifying red meat as a healthy animal food and assigning positive scores may be controversial. After testing, whether we assign positive or negative scores to red meat to calculate the aPDI, it does not change the conclusions of the study. The main reasons for our categorization were related to the following factors. Firstly, the participants included in this study were mainly farmers and herders, who have a high intensity of daily work and require a large amount of energy consumption, and poultry and red meat could provide a large amount of energy for their daily work. Secondly, the intake of poultry and red meat of the study participants was not higher than the maximum intake recommended by the Dietary Guidelines for Chinese Residents 2022; therefore, we considered that the participants’ intake of red meat could just meet their daily work consumption and there was no excessive intake. Finally, the red meat in this study was fresh beef and lamb naturally raised by local farmers and herders, excluding processed meat. Since our proposed aPDI assesses the overall dietary intake of the study population and does not emphasize the health effects of a single food group, the classification of foods takes into account factors such as the characteristics of the study population and the intake of food groups, rather than completely relying on the results of previous studies. The results of numerous studies support that animal fat, processed meat, and other animal foods have adverse effects on human health [40,41,43]. Therefore, this study defined animal fat, processed meat, and other animal foods as unhealthy animal foods.

Our findings suggest that high PDI and aPDI are related to a lower risk of dyslipidemia. In the PDI and dyslipidemia association analysis, Q3 and Q4 were protective factors for dyslipidemia compared to Q1, while Q5 was not associated with dyslipidemia, possibly because the PDI assessed overall plant-based food consumption, including unhealthy plant-based foods such as refined grains and sugary beverages. Not all plant foods are beneficial for individuals with dyslipidemia, and unhealthy plant foods may have a negative impact on lipid levels. Therefore, the results of this study suggest that an overall moderate and higher intake of plant foods may be preferable and the highest intake of plant foods may not reduce the risk of dyslipidemia. Plant foods are rich in a multitude of substances that are protective against dyslipidemia, such as dietary fiber, antioxidants, phytosterols, phenols, flavonoids, and sulfides, which can reduce blood lipid levels through a variety of mechanisms [44,45,46,47,48,49]. Meanwhile, the role of healthy animal foods included in aPDI on blood lipids cannot be ignored. Calcium intake from dairy products may improve blood lipid levels by inhibiting cholesterol reabsorption, restricting lipogenesis, or enhancing lipolysis [50,51]. Dairy products such as milk do not increase cholesterol as saturated fat [52]. Fish contains several types of unsaturated fatty acids, which are important for improving lipid profiles, especially LDL-C, HDL-C, and TG levels [53,54]. 

It is worth mentioning that PDI, hPDI, and aPDI were negatively correlated with low HDL-C in this study, while uPDI was positively correlated with low HDL-C, indicating that plant foods have significant effects on low HDL-C. Therefore, plant-based dietary interventions may be effective in the prevention of low HDL-C. However, the results of a previous meta-analysis showed that a vegetarian diet was associated with decreased HDL-C [11,12]. The discrepancy between studies may be due to the fact that vegetarian diets exclude all animal foods and do not distinguish between healthy and unhealthy plant-based foods. 

Our study had some strengths. For example, our study (1) involved a multi-ethnic region of Xinjiang, China, with distinct dietary characteristics, (2) considered the health effects of animal foods, and (3) provided information on the dietary patterns of a Chinese population. However, our study had a few limitations. First, the cross-sectional design could not determine the causal relationship between plant foods and dyslipidemia. Second, even though we controlled several multiple covariates, potential confounders may still exist. Third, the classification of animal foods was based on the available evidence and the characteristics of this study population, which may be subjective and therefore limited in extrapolation. Future validation in large sample populations based on scientific research evidence is still needed to continuously adjust and optimize the classification of animal foods.

## 5. Conclusions

In this multi-ethnic population study in Xinjiang, high PDI and aPDI were related to a lower risk of dyslipidemia. A reduction in the intake of unhealthy plant-based foods and animal-based foods may reduce the risk of dyslipidemia. High-quality plant-based dietary patterns may prevent the risk of low HDL-C. Follow-up data from cohort studies and clinical trials should be used to further optimize the classification of food groups based on population characteristics and to investigate potential mechanisms between plant-based foods and dyslipidemia, laying the foundation for more-intensive studies.

## Figures and Tables

**Figure 1 nutrients-15-00230-f001:**
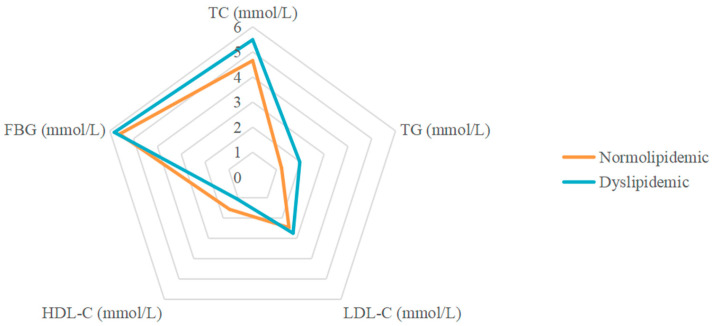
Radar plot of FPG and blood lipids in dyslipidemic and normolipidemic participants.

**Figure 2 nutrients-15-00230-f002:**
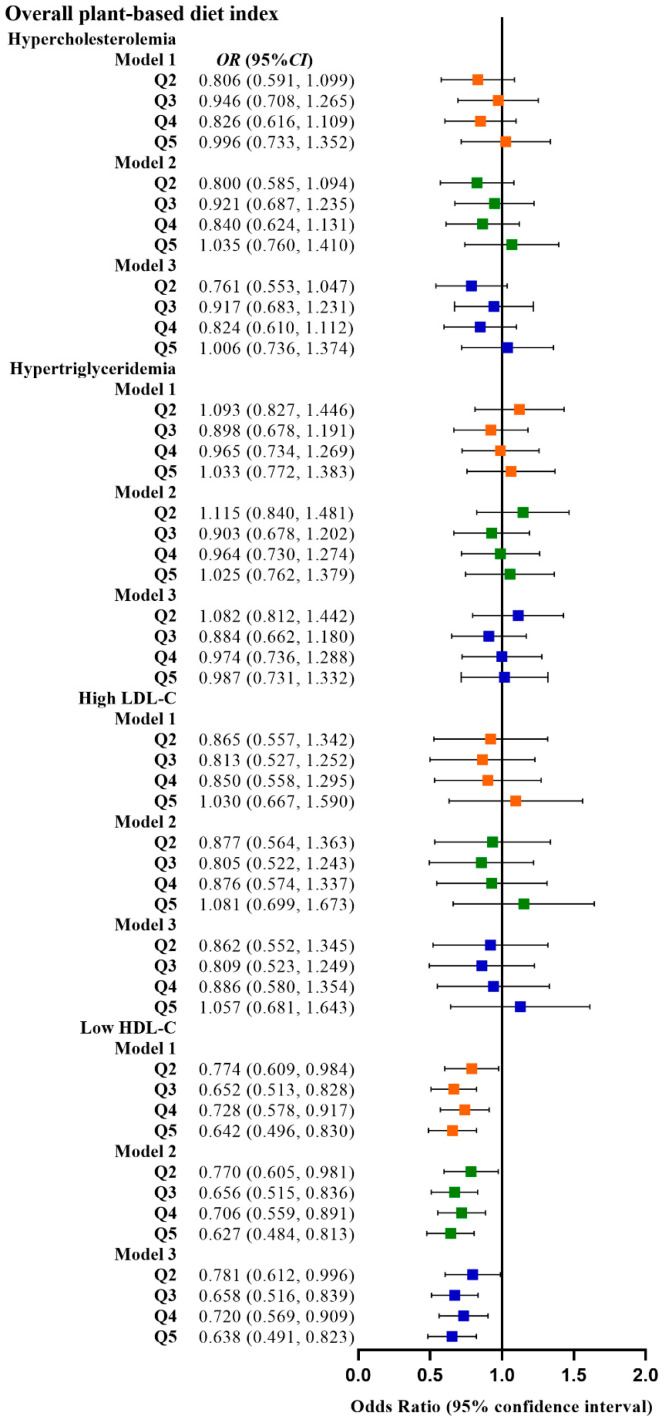
Odds ratios (OR) and 95% confidence intervals (CI) for PDI with different types of dyslipidemia. The squares show odds ratios and the line ranges show 95% confidence intervals. Model 1: unadjusted; model 2: adjusted for sex, age, ethnicity, BMI, socio-economic status, smoking status, alcohol consumption, and physical activity; model 3: additionally adjusted for T2DM and hypertension.

**Figure 3 nutrients-15-00230-f003:**
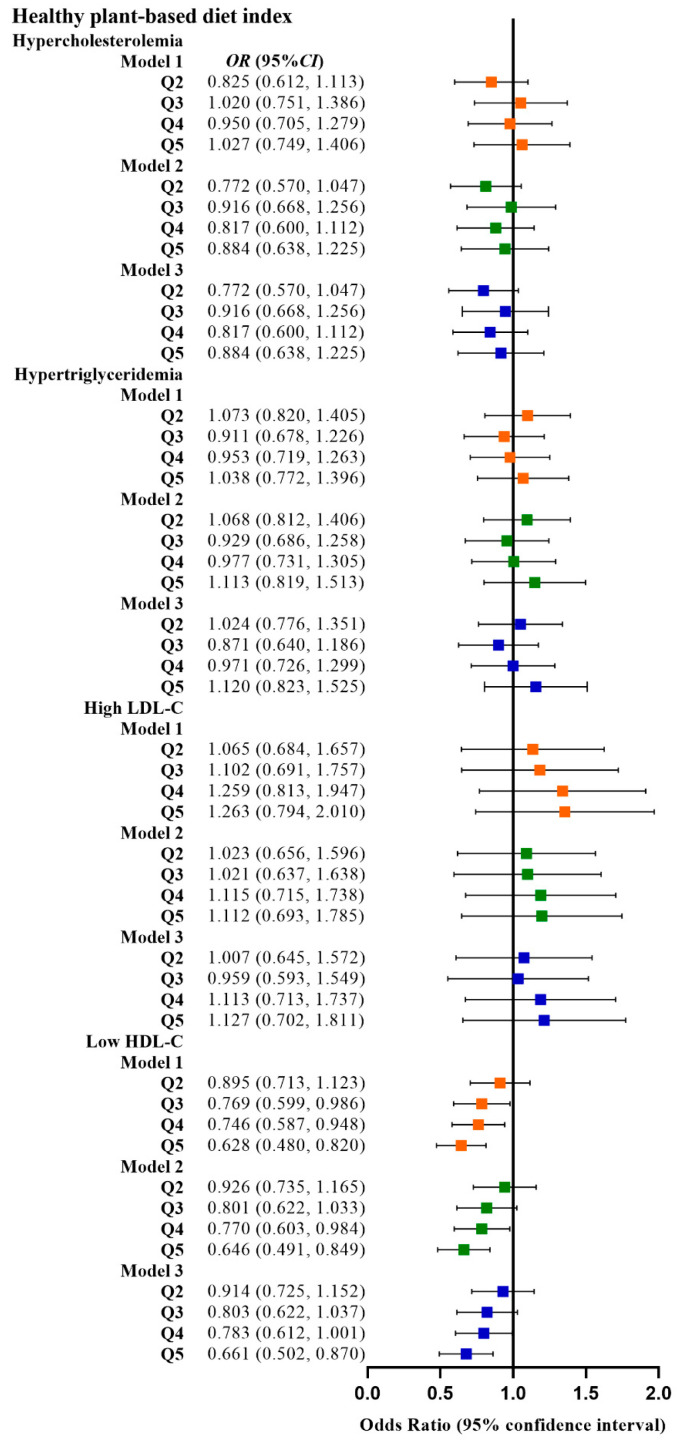
Odds ratios (OR) and 95% confidence intervals (CI) for hPDI with different types of dyslipidemia. The squares show odds ratios and the line ranges show 95% confidence intervals. Model 1: unadjusted; model 2: adjusted for sex, age, ethnicity, BMI, socio-economic status, smoking status, alcohol consumption, and physical activity; model 3: additionally adjusted for T2DM and hypertension.

**Figure 4 nutrients-15-00230-f004:**
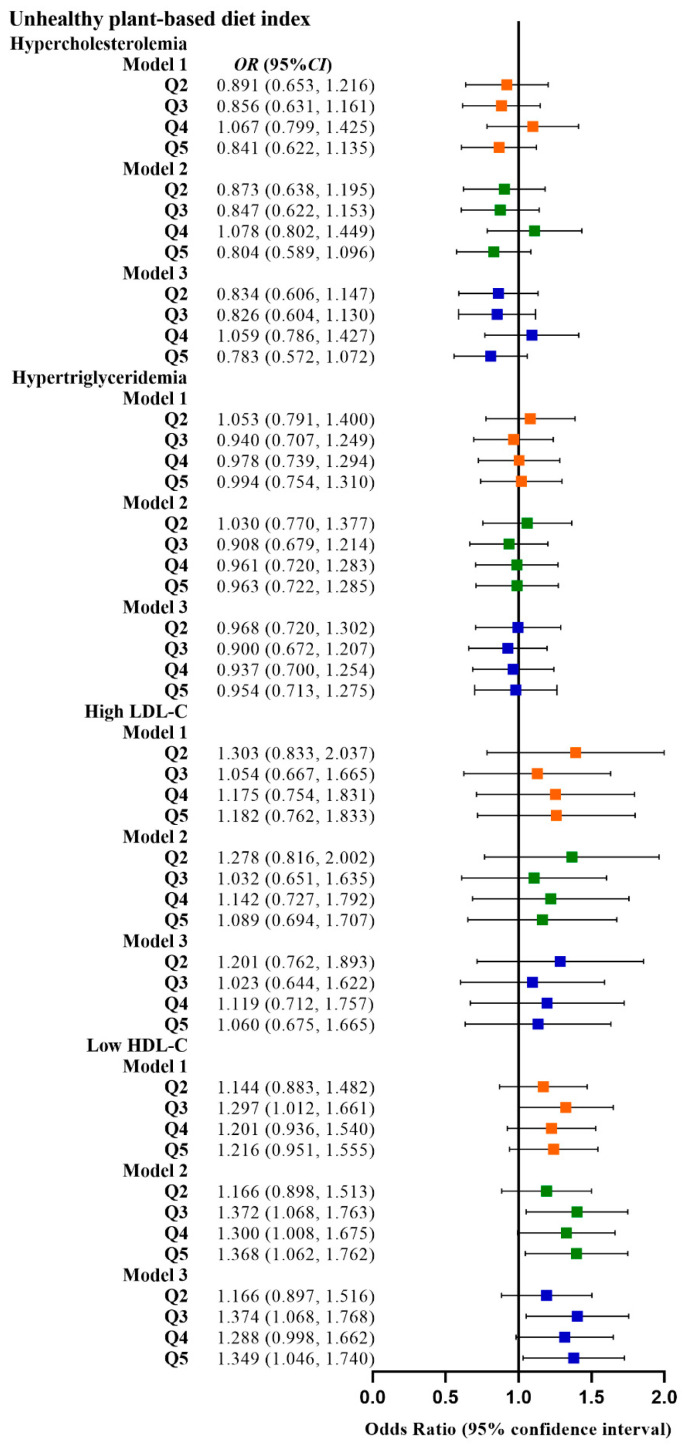
Odds ratios (OR) and 95% confidence intervals (CI) for uPDI with different types of dyslipidemia. The squares show odds ratios and the line ranges show 95% confidence intervals. Model 1: unadjusted; model 2: adjusted for sex, age, ethnicity, BMI, socio-economic status, smoking status, alcohol consumption, and physical activity; model 3: additionally adjusted for T2DM and hypertension.

**Figure 5 nutrients-15-00230-f005:**
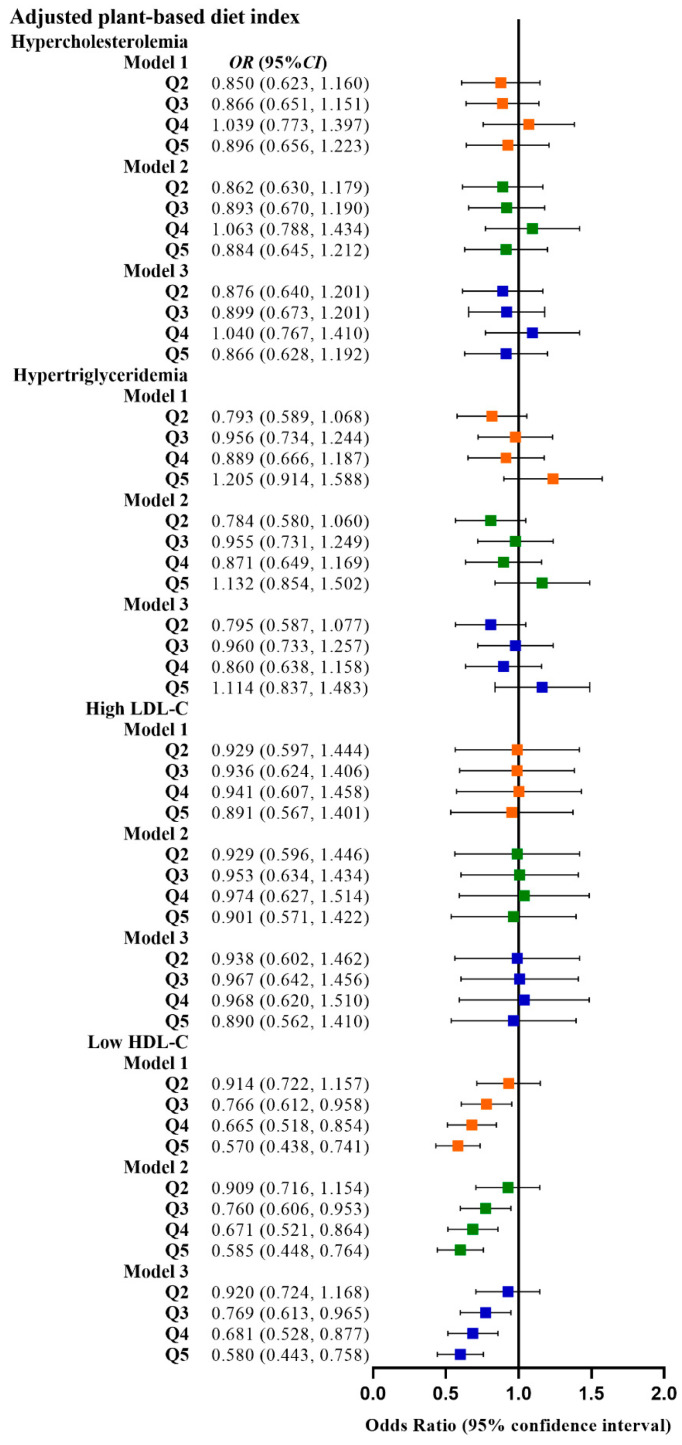
Odds ratios (OR) and 95% confidence intervals (CI) for aPDI with different types of dyslipidemia. The squares show odds ratios and the line ranges show 95% confidence intervals. Model 1: unadjusted; model 2: adjusted for sex, age, ethnicity, BMI, socio-economic status, smoking status, alcohol consumption, and physical activity; model 3: additionally adjusted for T2DM and hypertension.

**Table 1 nutrients-15-00230-t001:** Demographic and clinical characteristics of study participants.

	Dyslipidemic Participants (n = 1501)	Normolipidemic Participants (n = 2595)	*p*-Value
Sex, n (%)			0.010
Male	714 (47.6)	1126 (43.4)	
Female	787 (52.4)	1469 (56.6)	
Age (years)	51.42 ± 10.16	51.04 ± 10.19	0.247
Ethnicity, n (%)			<0.001
Han	182 (12.1)	337 (13.0)	
Hui	516 (34.4)	745 (28.7)	
Uygur	282 (18.8)	914 (35.2)	
Kazak	489 (32.6)	550 (21.2)	
other	32 (2.1)	49 (1.9)	
Body mass index (kg/m^2^)	26.80 ± 4.14	26.14 ± 4.30	<0.001
Socio-economic status, n (%)			0.254
Low	846 (56.4)	1451 (55.9)	
Medium	619 (41.2)	1100 (42.4)	
High	36 (2.4)	44 (1.7)	
Marriage			0.519
Married	1331 (88.7)	2318 (89.3)	
Other	170 (11.3)	277 (10.7)	
Smoking status, n (%)			0.114
Never	1135 (75.6)	2030 (78.2)	
Occasionally	30 (2.0)	55 (2.1)	
Every day	336 (22.4)	510 (19.7)	
Alcohol drinking, n (%)			0.322
Never	1250 (83.3)	2207 (85.0)	
Occasionally	230 (15.3)	355 (13.7)	
Weekly	21 (1.4)	33 (1.3)	
Physical activity, n (%)			0.531
Never	1237 (82.4)	2119 (81.7)	
Occasionally	211 (14.1)	366 (14.1)	
Weekly	53 (3.5)	110 (4.2)	
FPG (mmol/L)	5.81 ± 2.05	5.59 ± 1.40	<0.001
TC (mmol/L)	5.49 ± 1.66	4.65 ± 0.79	<0.001
TG (mmol/L)	1.98 ± 1.48	1.21 ± 0.46	<0.001
LDL-C (mmol/L)	2.75 ± 1.37	2.47 ± 0.53	<0.001
HDL-C (mmol/L)	1.06 ± 0.90	1.57 ± 0.49	<0.001
TC/HDL-C (*M* (*P_25_*, *P_75_*))	4.89 (3.87, 15.50)	3.09 (2.56, 3.59)	<0.001
LDL-C/HDL-C ((*P_25_*, *P_75_*))	2.62 (1.98, 5.58)	1.66 (1.32, 1.95)	<0.001
AIP (*M* (*P_25_*, *P_75_*))	0.33 (0.06, 0.62)	−0.11 (−0.28, 0.03)	<0.001

Data are expressed as the mean (SD) or n (%). *p*-values were obtained from the *t*-test/Mann–Whitney U test for continuous variables and Chi-square test for categorical variables.

**Table 2 nutrients-15-00230-t002:** Multivariate logistic regression analyses showing associations between dyslipidemia and quintiles of plant-based diet indices.

	Quintile 1		Quintile 2		Quintile 3		Quintile 4		Quintile 5	
	*OR* (95%*CI*)	*p*	*OR* (95%*CI*)	*p*	*OR* (95%*CI*)	*p*	*OR* (95%*CI*)	*p*	*OR* (95%*CI*)	*p*
Overall plant-based diet index										
Model 1	1 (Reference)		0.865 (0.710, 1.055)	0.152	0.787 (0.648, 0.955)	0.015	0.801 (0.662, 0.969)	0.022	0.835 (0.680, 1.025)	0.085
Model 2	1 (Reference)		0.864 (0.708, 1.055)	0.152	0.784 (0.645, 0.953)	0.015	0.794 (0.655, 0.962)	0.018	0.832 (0.676, 1.023)	0.080
Model 3	1 (Reference)		0.855 (0.699, 1.046)	0.128	0.780 (0.641, 0.949)	0.013	0.799 (0.659, 0.970)	0.023	0.825 (0.670, 1.017)	0.071
Healthful plant-based diet index										
Model 1	1 (Reference)		0.961 (0.795, 1.162)	0.681	0.874 (0.713, 1.072)	0.197	0.906 (0.746, 1.102)	0.324	0.894 (0.725, 1.102)	0.293
Model 2	1 (Reference)		0.964 (0.796, 1.169)	0.712	0.885 (0.719, 1.089)	0.248	0.899 (0.737, 1.098)	0.297	0.890 (0.718, 1.103)	0.288
Model 3	1 (Reference)		0.941 (0.776, 1.142)	0.540	0.867 (0.703, 1.069)	0.181	0.893 (0.731, 1.092)	0.272	0.893 (0.720, 1.109)	0.306
Unhealthful plant-based diet index										
Model 1	1 (Reference)		1.068 (0.871, 1.309)	0.528	1.047 (0.859, 1.277)	0.649	1.187 (0.976, 1.444)	0.086	1.019 (0.838, 1.240)	0.847
Model 2	1 (Reference)		1.067 (0.869, 1.310)	0.536	1.068 (0.873, 1.305)	0.524	1.233 (1.010, 1.505)	0.039	1.065 (0.871, 1.302)	0.541
Model 3	1 (Reference)		1.036 (0.842, 1.275)	0.736	1.058 (0.864, 1.296)	0.584	1.208 (0.988, 1.477)	0.065	1.042 (0.851, 1.277)	0.689
Adjusted plant-based diet index										
Model 1	1 (Reference)		0.836 (0.684, 1.020)	0.077	0.864 (0.719, 1.038)	0.119	0.760 (0.622, 0.929)	0.007	0.752 (0.614, 0.923)	0.006
Model 2	1 (Reference)		0.838 (0.686, 1.025)	0.085	0.873 (0.725, 1.050)	0.149	0.771 (0.630, 0.944)	0.012	0.752 (0.612, 0.925)	0.007
Model 3	1 (Reference)		0.854 (0.697, 1.045)	0.125	0.881 (0.731, 1.061)	0.182	0.770 (0.628, 0.945)	0.012	0.748 (0.607, 0.921)	0.006

Model 1: unadjusted. Model 2: adjusted for sex (men/women), age (continuous), ethnicity (Han/Uygur/Hui/Kazak/other), BMI (continuous), socio-economic status (low/medium/high), smoking status (never/occasionally/every day), alcohol consumption (never/occasionally/every day), and physical activity (never/occasionally/weekly). Model 3: model 2 plus T2DM (yes/no) and hypertension (yes/no).

## Data Availability

Data are available on request due to restrictions, e.g., privacy or ethical.

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
