# Peer review of "Association between Different Types of Plant-Based Diets and Dyslipidemia in Middle-Aged and Elderly Chinese Participants"

_nutrients, 2023, doi:10.3390/nu15010230_

Round 1
Reviewer 1 Report
This study analyzed the relationship between plant-based diets and the risk of dyslipidemia in middle-aged and elderly population from Xinjiang, China. The topic is important and the manuscript concludes that the quality of plant foods is important in preventing dyslipidemia as a risk factor of cardiovascular diseases and stroke. I would recommend this manuscript for publication after the following suggestions have been attended to:
Lines 138-139: what was the reason to include red meat between healthy animal foods? Do you think this explain part of the outcomes?
Line 239 (Discussion): what could be the explanation that Q3 and Q4 (PDI) were associated with lower odds of dyslipidemia but not Q5?
Lines 298-320: in this paragraph, one important idea can be added - plant bioactive compounds exert their antioxidant activity through multiple mechanisms, including the activation of the Nrf2/ARE pathway and down-regulation of the NF-kB pathway, directly implicated in the inflammatory response ((doi: 10.3390/antiox11071412). This recent meta-analysis of RCTs in middle-aged and older adults revealed that walnut-enriched diets significantly lowered triglyceride, total cholesterol, and LDL cholesterol concentration levels, with no consequences on anthropometric and glycemic parameters
Lines 331-339: another limitation - in cohort studies, when the incidence of events is high, OR can overstate the size of the effect (doi: 10.1503/cmaj.101715)
References should be checked
Author Response
Dear Reviewers:
We sincerely thank the editor and all reviewers for their valuable comments on our manuscript entitled " Association between different types of plant based diets and dyslipidemia in middle-aged and elderly Chinese participants ". Those comments are highly insightful and constructive, and enabled us to greatly improve the quality of our manuscript. We have addressed the comments very carefully. All the changes made in the revised manuscript are highlighted in yellow.
We shall look forward to hearing from you at your earliest convenience.
Kind regards,
Professor Jianghong Dai, PhD
Department of Epidemiology and Biostatistics,
School of Public Health Xinjiang Medical University.
Urumqi, 830001, China
Email: [email protected]

Reviewer 2 Report
This is an interesting study of Wang et al, investigating the possible associations between plant-based diets and dyslipidemia. Τhe topic is of great interest especially in light of recent findings regarding the benefits of such diets. However, there are some important issues that need to be addressed by the authors.
· Was the food frequency questionnaire validated? If so, a relevant reference should be added.
· Line 118: It is not clear if the methodology for calculating PDI, hPDI, and uPDI has been previously reported and validated or if it is a new methodology lunched by the authors. The 2 references mentioned are metanalyses and they do not describe these indexes, their scoring system or their validation. This is extremely important as the authors have built the whole analysis on these indexes and it must be clarified and adequately supported by the appropriate references.
· Line 138: What is the justification for including red meat in the healthy animal food category?
· The word gender should be replaced with sex throughout the results, discussion, and tables.
· There must be an additional model in the statistical analysis adjusting also for history of chronic diseases, as many such diseases (i.e. diabetes, hypertension) are associated with dyslipidemia.
· The figures contain too much information. The authors should consider presenting only the results of the highest quantile.
· The conclusion that healthy plant-based diet is associated with lower risk of dyslipidemia is overstated as the main findings of the study relate to lower risk of low HDL
· The manuscript would benefit from adding as outcomes more dyslipidemia indexes like Total Cholesterol to HDL ratio, LDL to HDL ratio, and Atherogenic Index of Plasma (AIP) calculated as log10 (TG/HDL-C).
Author Response

(The authors gave the same response as above.)

Round 2
Reviewer 2 Report
Dear authors
I believe there has been a substantial revision to the submitted manuscript which has enabled better understanding and presentation of the results. However, there are still a few issues that need to be addressed.
1. Table 2 It is not clear what dyslipidemia refer to. Is it high total cholesterol? Is it low HDL cholesterol? Something else? Why did the authors choose to present only one type of dyslipidemia in this table?
2. In my previous review report I suggested that you could use more dyslipidemia indices as outcomes. I see that you used AIX in the description of the study group characteristics in Table 1, but I cannot see the other two suggested indices and most importantly I cannot see any of them used as outcomes. In your response you say that AiX might be found in Table 2 but I cannot see that. It would be good to present the associations of all types of dyslipidemia including the suggested indices with the types of plant based diets even as supplemental material
Author Response
Dear Editor and Reviewers:
We sincerely thank the editor and all reviewers for their valuable comments
on our manuscript entitled " Association between different types of
plant-based diets and dyslipidemia in middle-aged and elderly Chinese
participants ". Those comments are highly insightful and constructive, and
enabled us to greatly improve the quality of our manuscript. We have
addressed the comments very carefully. All the changes made in the revised
manuscript are highlighted in yellow. The revisions in the manuscript and the
responses to the reviewers' comments are listed below.
